# Colorimetric Approach for Nucleic Acid *Salmonella* spp. Detection: A Systematic Review

**DOI:** 10.3390/ijerph191710570

**Published:** 2022-08-25

**Authors:** Asma Nadia Ahmad Faris, Mohamad Ahmad Najib, Muhammad Najmi Mohd Nazri, Amir Syahir Amir Hamzah, Ismail Aziah, Nik Yusnoraini Yusof, Rohimah Mohamud, Irneza Ismail, Fatin Hamimi Mustafa

**Affiliations:** 1Institute for Research in Molecular Medicine (INFORMM), Health Campus, Universiti Sains Malaysia, Kubang Kerian 16150, Kelantan, Malaysia; 2Faculty of Biotechnology and Biomolecular Sciences, Universiti Putra Malaysia, Serdang 43400, Selangor, Malaysia; 3Department of Immunology, School of Medical Sciences, Universiti Sains Malaysia, Kubang Kerian 16150, Kelantan, Malaysia; 4Advanced Devices & System (ADS) Research Group, Department of Electrical & Electronic Engineering, Faculty of Engineering and Built Environment, Universiti Sains Islam Malaysia, Bandar Baru Nilai, Nilai 71800, Negeri Sembilan, Malaysia

**Keywords:** colorimetric, nucleic acid, *Salmonella*, detection, systematic review

## Abstract

Water- and food-related health issues have received a lot of attention recently because food-poisoning bacteria, in particular, are becoming serious threats to human health. Currently, techniques used to detect these bacteria are time-consuming and laborious. To overcome these challenges, the colorimetric strategy is attractive because it provides simple, rapid and accurate sensing for the detection of *Salmonella* spp. bacteria. The aim of this study is to review the progress regarding the colorimetric method of nucleic acid for *Salmonella* detection. A literature search was conducted using three databases (PubMed, Scopus and ScienceDirect). Of the 88 studies identified in our search, 15 were included for further analysis. *Salmonella* bacteria from different species, such as *S.* Typhimurium, *S.* Enteritidis, *S.* Typhi and *S.* Paratyphi A, were identified using the colorimetric method. The limit of detection (LoD) was evaluated in two types of concentrations, which were colony-forming unit (CFU) and CFU per mL. The majority of the studies used spiked samples (53%) rather than real samples (33%) to determine the LoDs. More research is needed to assess the sensitivity and specificity of colorimetric nucleic acid in bacterial detection, as well as its potential use in routine diagnosis.

## 1. Introduction

Salmonellosis ranges clinically from common *Salmonella* gastroenteritis (diarrhea, stomach cramps and fever) to enteric fevers (including typhoid fever). *Salmonella* Typhi and Paratyphi A, B and C cause enteric fever, a potentially severe and life-threatening febrile infection. Every year, an estimated 11.9 million to 27.1 million cases of enteric fever are reported around the world [1,2]. According to a systematic analysis on typhoid and paratyphoid, 10.9 million cases of typhoid fever and 3.4 million cases of paratyphoid fever were reported worldwide in 2017, with 116.8 thousand deaths and 19.1 thousand deaths, respectively [3]. This disease is widespread in South and Central Asia, Africa, and South America, with fatality rates ranging from 10 to 100 per 100,000 people each year. The Ministry of Health Malaysia revealed that the incidence rate of typhoid and paratyphoid fever in Malaysia in 2020 was 0.55, and the fatality rate was 0.02 per 100,000 inhabitants [4]. Due to the lack of uncontaminated drinking water and other domestic reasons, outbreaks in Malaysia occur in specific parts of the country, such as in the state of Kelantan and Sabah [4]. In Kelantan, there was a large outbreak in 2005, with 888 cases and two deaths reported. The unexpected rise in cases could be linked to the flood in late 2004, contaminated ice and contaminated food provided by unvaccinated street hawkers [5].

Humans are the sole natural hosts for *S.* Typhi and *S.* Paratyphi A, B and C. One of the common transmissions is via the unwashed hands of food workers which contaminate foods. *Salmonella* spp. is mostly distributed by the fecal–oral route, although it can also be disseminated through food and drink, as well as direct contact with animals and people [6]. Animal-based foods, such as pork, chicken, beef, milk and eggs, are frequently contaminated; thus, ensuring meat is well cooked is essential. *Salmonella* spp. can also be presented in the excrement of some dogs, which then affects humans who have direct contact with these animals [7]. *Salmonella* spp. enters the body by passing through the stomach and attaching to epithelial cells that line the small intestine. To trigger phagocytosis, the bacteria release toxins into the cells and then proliferate in phagosomes [7]. The bacteria kill the host cells, causing various symptoms including fever, diarrhea, abdominal cramps, nausea and vomiting. If gone untreated, the bacteria enter the bloodstream to affect organs such as the liver and bones [7].

Bacterial culture using the samples of blood and stools is currently used in the laboratory to diagnose typhoid and paratyphoid fever, and this is regarded the gold standard for diagnosis. Another technique to identify the presence of *Salmonella* bacteria is polymerase chain reaction (PCR) associated with agarose gel electrophoretic analysis. However, these methods (bacterial culture and PCR) are lab-based, need highly trained workers and also require several days for the isolation and identification of the causative agents [8]. Rapid detection methods such as Surface Plasmon Resonance (SPR) enable the sensitive, label-free and real-time monitoring of *Salmonella* conservative genes by fixing ssDNA on the surface of a gold plate to create a SPR nucleic acid sensor [9]. However, such sensitivity has significant limitations in actual detections, making it difficult to quickly identify *Salmonella* at low concentrations [10]. Furthermore, the use of electrochemical immunosensors is also one of the rapid detection methods of *Salmonella* in foods, but this method is complicated and requires skilled technicians [11]. Thus, a rapid and easy-to-use approach for the detection of *Salmonella* spp. would aid in clinical diagnosis and further lead to disease prevention. The colorimetric method is to measure the concentration of a chemical element or compound in a solution with the aid of a color reagent [12]. Comparing variations in color and its intensity will yield colorimetric signals. Commonly, pH-sensitive dyes and fluorescent probes are used to identify nucleic acids. Recently, colorimetric nucleic acid detection with the use of noble metal nanoparticles such as gold has attracted interest [13]. Gold nanoparticles (AuNPs) exhibit unique optical properties which determine the color of AuNP solutions and their absorbance in the UV-vis region used for the colorimetric analysis of biological samples [12].

Currently, several nucleic-acid-based detection assays for *Salmonella* detection have been developed, such as loop-mediated isothermal amplification (LAMP) and the enzyme-linked oligonucleotide assay (ELONA) [14,15]. However, the advancements of the developed assays have not been well documented in a comprehensive review. Such a review is needed to highlight the strategies deployed, their performance, research gaps and the limitations of the assays. Therefore, this review aims to comprehensively summarize the use of colorimetric nucleic acid detection assays for *Salmonella* spp. detection.

## 2. Materials and Methods

This systematic review utilized the preferred reporting items for systematic review and meta-analyses (PRISMA) guidelines. The protocol of this systematic review and meta-analysis was registered in the PROSPERO database (CRD42022320571).

### 2.1. Search Strategy

The literature search was conducted in February 2022. The search was conducted through three databases (PubMed, Scopus and ScienceDirect) using lists of keywords. These keywords were combined using the Boolean operator AND (between key concepts) as follows: [Colorimetric] AND [Nucleic Acid] AND [*Salmonella*] AND [Detection]. An additional search was conducted by manually screening the references in the retrieved literature.

### 2.2. Selection of Studies

Articles were excluded if (i) the studies did not involve the development and evaluation of nucleic-acid-based colorimetric detection; (ii) the studies were published in languages other than English or Malay; (iii) the studies detected pathogens other than *Salmonella* spp. The retrieved literature was downloaded into Endnote reference manager, and duplicates were identified and removed. The references were distributed to two authors, who independently reviewed the titles and abstracts. Satisfactory agreement regarding the screening process was reached between the authors. The two authors performed full-text screening and summarized the findings.

### 2.3. Data Extraction

Data from the selected sources were collated and summarized using a standard charting table consisting of eight domains: (i) type of bacteria; (ii) reporters; (iii) food samples; (iv) type of sample; (v) limit of detection (LoD); (vi) sensitivity; (vii) specificity; (viii) year of publication.

### 2.4. Assessment of the Quality of Studies

The quality of the included studies was assessed using the Critical Appraisal Skills Programme (CASP) Qualitative Checklist. The CASP Qualitative Checklist comprises seven questions as follows: 1. Was there a clear statement of the aims of the research? 2. Was the research design appropriate to address the aims of the research? 3. Was the execution of the index test described in sufficient detail to permit replication of the test? 4. Did the study provide a clear definition of what was considered to be a positive result? 5. If necessary, have ethical issues been taken into consideration? 6. Was the data analysis sufficiently rigorous? 7. Is there a clear statement of findings? The assessment was performed by three authors independently, who judged the quality of the studies using the answers “no”, “yes”, “unclear” or “not applicable”. If there were disagreements among the authors, the disagreements were resolved by discussion.

## 3. Results

### 3.1. Retrieved Articles

Of the 88 studies that were identified from three databases, 60 remained after 28 duplicates were removed. Of these, 38 were excluded during title and abstract screening, and 7 were excluded during full-text screening based on the study criteria, meaning a total of 45 studies that were excluded. Fifteen studies were included in the final review (Figure 1). A total of 15 articles were analyzed to extract data for further evaluation (Table 1).

### 3.2. Quality of the Selected Articles

A summary of the CASP Qualitative Checklist assessment is presented in Figure 2. The overall results of the quality assessment showed a low risk of bias in all 15 studies. Regarding the answers for each quality question, all the articles clearly stated the aims of the research, but around 7% of the articles did not show an appropriate experimental design. All the articles (100%) explained the index test in detail. In the analysis of the definition of a positive result in testing diagnostics, all the articles (100%) reported a cut off value. A total of 13% of the articles unclearly stated the ethical issues, while 7% did not consider the ethical issues when biological samples were used. All articles (100%) reported sufficiently rigorous analyses of their data and displayed their findings.

### 3.3. Types of Salmonella Species

All 15 studies reported the development of nucleic-acid-based diagnostics for the detection of *Salmonella* bacteria. The diagnostic platform detected *Salmonella* bacteria from five different species, as shown in Figure 3. The discovered bacteria in the retrieved studies came from the species *S.* Typhimurium (53%), *S.* Enteritidis (40%), *Salmonella* spp. (27%), *S.* Typhi (7%) and *S.* Paratyphi A (7%).

### 3.4. Type of Sample

The majority of research (73%) employed nucleic acid as a sample or target, while 27% used a bacteria suspension, as shown in Figure 4.

### 3.5. Performance of Nucleic-Acid-Based Colorimetric Assays

With regard to the analytical sensitivity of the nucleic-acid-based colorimetric detection, the limit of detection (LoD) was presented in two types of concentrations, which were CFU per mL and CFU. The majority of the studies (10 studies) reported the LoD in CFU/mL. Of CFU per mL unit, the detection of *Salmonella* Typhimurium and *Salmonella* Enteritidis showed the lowest LoD out of all of the species, with the LoD of 2.56 CFU/mL [17]. Two studies did not report the LoD of the assays [23,27]. In order to determine the LoD of the samples, the majority of studies used spiked samples (53%) rather than real samples (33%). Two studies evaluated the LoD directly using a bacterial suspension without spiking on any sample [15,20].

Three out of fifteen studies reported on diagnostic sensitivity and specificity [18,22,24]. The studies mostly regarded *Salmonella* Typhimurium and *Salmonella* Enteritidis. The detection strategies or type of assays used for the detection of these species were aptamer-based colorimetric assays (*Salmonella* Typhimurium and *Salmonella* Enteritidis), PCA (*Salmonella* Enteritidis) and sandwich hybridization system assays (*Salmonella* spp.).

## 4. Discussion

Non-typhoidal salmonellosis refers to illnesses caused by all serotypes of *Salmonella* except for Typhi, Paratyphi A, Paratyphi B and Paratyphi C. Non-typhoidal *Salmonella* includes strains such as *S.* Typhimurium and *S.* Enteritidis. Typhoid fever and paratyphoid fever are still major public health problems that cause significant morbidity and mortality worldwide, especially in developing and underdeveloped countries. The effective management of these diseases depends on the performance and turnaround time of diagnostic tests. Diagnostic techniques based on nucleic acid colorimetric detection have recently attracted the attention of researchers due to their excellent accuracy. Colorimetric analysis is the method of using a color reagent to determine the concentration of a chemical element or chemical compound in a solution. This type of analysis only requires a short assay time; the colorimetric responses can be observed in 30 min to 2 h [30], and the results can be verified with the naked eye. The whole colorimetric experiment is sensitive, simple and straightforward, and it also does not require complicated instruments, which makes it is one of the best approaches for the rapid in-field detection of *Salmonella*, especially in food safety screening [31]. It is an important component of food quality and safety protocols [32]. This colorimetric method not only detects *Salmonella* spp. but also other bacteria such as *Shigella*, *Vibrio cholera, Listeria monocytogenes and Escherichia coli* [15,18]. Regarding the 15 studies included in the final review, in most of the studies, *S.* Typhimurium and *S.* Enteritidis were detected. This is because these are the most common strains of *Salmonella* that can be found in food supplies, and they are the most common serotypes associated with foodborne illness. In 2017, a total of 535,000 non-typhoidal *Salmonella* invasive disease infections and 77,500 deaths were reported [33]. Non-typhoidal *Salmonella* infection usually results in self-limiting gastroenteritis; however, dehydrated patients may require hospitalization. Despite the fact that typhoid and paratyphoid fever generate more than 25 times the number of cases as non-typhoidal *Salmonella* invasive disease, the number of deaths caused by each disease is comparable [33]. There is an urgent need for future research to focus on *S.* Typhi and *S.* Paratyphi, as these typhoidal *Salmonella* are major public health problems.

Most studies used AuNPs as a reporter particle for the colorimetric detection of nucleic acid. Because of their ease of use, cost-effective manufacturing and convenience of use, AuNPs have been widely used in colorimetric assays. More importantly, the AuNP sensor response is a visual color shift, making the data easy to interpret [34]. The aggregation of AuNPs results in a shift in the absorption wavelength and a color change from red to blue. AuNP-colorimetric sensing strategy is quick and sensitive, and it also has several uses in real-time on-site monitoring and in rapid tests of food quality and safety [32]. Nanoparticles have limitations such as high cost for scale production [13]. Some studies use aptamers as a substitute choice to overcome this challenge. Aptamers are oligonucleotides that can be obtained through SELEX (systematic evolution of ligands by exponential enrichment). Aptamers can specifically bind to a target molecule and produce a unique structure, comparable to nanoparticles. Furthermore, aptamers are important in biotechnology because they are affordable and stable under a variety of experimental circumstances [13]. Aptamers seem to be promising alternatives as capturing agents.

All studies used food samples. Most of the food samples were inoculated with nucleic acid and then underwent DNA extraction and amplification before detection. The amplification method that was used in these studies was Polymerase Chain Reaction (PCR) to improve the sensitivity and colorimetric signal. If a target in a sample is amplified using PCR, it will improve the test’s sensitivity to detect the target, and also, the LoD will decrease after PCR. The lower the LoD, the better the sensitivity. Real samples are used to evaluate a test’s diagnostic accuracy (sensitivity and specificity), while spiked samples are utilized to see if analyte detection is affected by the biological sample matrix and to validate recovery. In future studies, testing on clinical samples such as blood and stool samples is essential. The detection of pathogenic bacteria that pose a significant risk to human health necessitates a method that is rapid, reliable, convenient, sensitive and low-cost, all of which are the characteristics of the colorimetric method. Another significant advantage of the colorimetric approach is the ability of its results to be qualitatively or semi-qualitatively detected by the naked eye [35], as this visual detection (with the naked eye) is most suitable for point-of-care testing. Most assays are developed for use in the food industry by using preservation methods that prevent microbiological or biochemical changes; it is also possible to lengthen the time that a food remains healthy (its shelf life), giving more time for distribution, sales and home storage.

In this systematic review, there are a few studies that determined sensitivity and specificity. Sensitivity and specificity play an important role in diagnostic applications. Sensitivity indicates a diagnostic method’s ability to detect infection. The more sensitive a test is, the fewer false negative results there are; thus, this helps to detect infection. Specificity is a diagnostic method’s ability to detect individuals who do not have a disease. Thus, the better the specificity, the fewer false positive results there are [36]. This review found that the level of progress of studies involving the development of colorimetric nucleic acid detection for *Salmonella* still lags behind. This is because most studies only evaluated the LoD, not sensitivity and specificity; only 1/5 of the studies reported on sensitivity and specificity. The higher sensitivity (100%) and specificity (96%) of [18] was observed in the studies. They reported that this higher percentage was due to the use of reporters such as AuNPs and the use of small sample sizes compared to the other two studies which used RPA and DNA probes [22,24]. This suggests that reporters such as AuNPs are also promising nanoparticles that can be used to enhance the sensitivity and specificity of *Salmonella* species detection. Future studies should focus on evaluating sensitivity and specificity to ensure the accuracy of a test. Such an evaluation will confirm whether or not cross-reactivity may occur by ruling out false positive results [37].

There are some limitations to this review. We could not include meta-analyses since not enough studies have used the colorimetric nucleic acid detection method to evaluate sensitivity and specificity. Therefore, it is important to determine the sensitivity and specificity in diagnostic studies to evaluate effectiveness. Furthermore, a few studies did not mention serotypes of *Salmonella*, which made it difficult for us to separate them according to their species. The limitation of the colorimetric method is that background color of food samples may interfere with the visualization of the results, which will affect the test’s accuracy [38]. Another disadvantage of this method is that colorless compounds cannot be analyzed. For example, the enzyme-linked immunosorbent assay (ELISA) requires additional reagents such as streptavidin-HRP, TMB or hydrochloric acid (HCL) to produce colors [39].

## 5. Conclusions

The present systematic review provided an overview of the performance of diagnostic tests for typhoidal and non-typhoidal diseases. Varied LoDs were obtained using spiked samples ranging from 2.56 CFU/mL to 5 × 10^3^ CFU/mL. Different *Salmonella* bacteria were detected, including *S.* Typhimurium, *S.* Enteritidis, *S.* Typhi and *S.* Paratyphi A. Most of the studies used gold nanoparticles as the reporter to provide visible results for analysis. Nucleic acid is now being used in more studies since it can improve a test’s sensitivity through the amplification method. When compared to antigen detection, however, this approach can only detect bacterial antigens in a sample, limiting its diagnostic sensitivity. This review identified that there is a lack of studies that evaluate clinical samples using this colorimetric nucleic acid technique for bacterial detection. As a result, we recommend that more research should be conducted regarding clinical bacterial samples’ detection rather than spiked samples, and LoD sensitivity and specificity parameters should be included in these studies.

## Figures and Tables

**Figure 1 ijerph-19-10570-f001:**
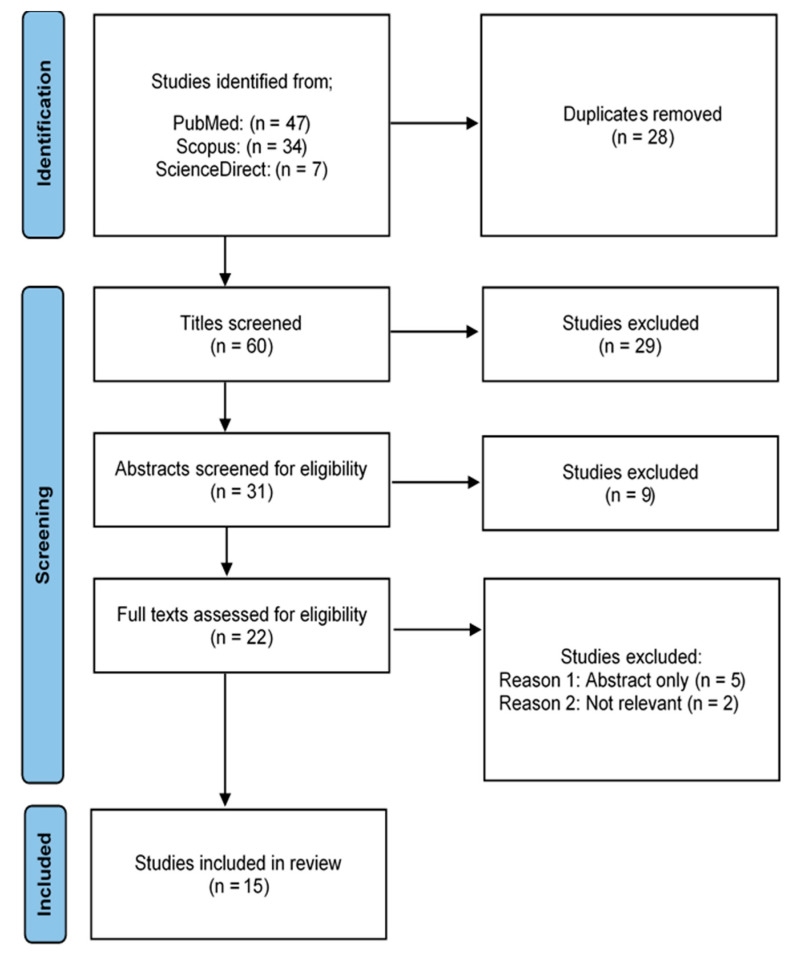
Study flow diagram during the selection process. n: number of studies.

**Figure 2 ijerph-19-10570-f002:**
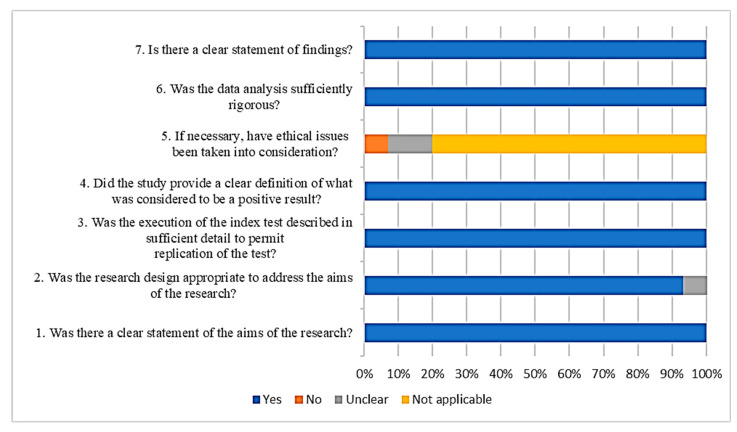
Quality assessment of the retrieved articles.

**Figure 3 ijerph-19-10570-f003:**
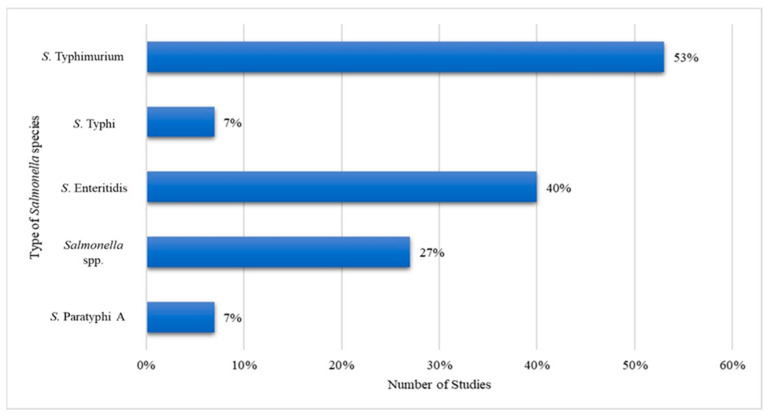
*Salmonella* bacteria from various species detected via nucleic-acid-based colorimetric method.

**Figure 4 ijerph-19-10570-f004:**
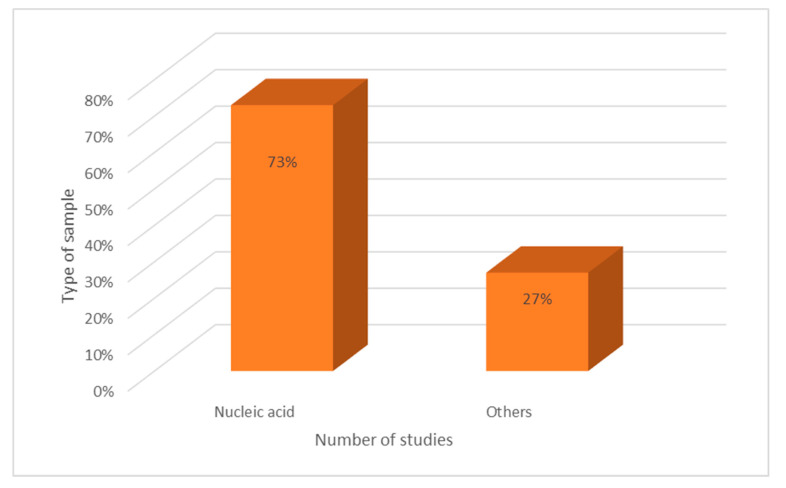
Types of samples used in nucleic-acid-based colorimetric detection.

**Table 1 ijerph-19-10570-t001:** Summary of the studies included.

No.	Type of Bacteria	Reporters	Food Samples	Type of Sample	Limit of Detection (LoD)	Sensitivity	Specificity	References
1	*Salmonella* Enteritidis, *Salmonella* Typhimurium	Aptamer-HRP-TMB	Milk sample (spiked)	Bacteria suspension	10^3^ CFU/mL	NR	NR	[16]
2	*Salmonella* Typhimurium, *Salmonella* Enteritidis	Gold nanoparticles	Lettuce sample (spiked)	Nucleic acid	2.56 CFU/mL	NR	NR	[17]
3	*Salmonella* Typhimurium, *Salmonella* Enteritidis	Gold nanoparticles	50 meat samples	Bacteria suspension	10^5^ CFU/mL	100%	96%	[18]
4	*Salmonella* Typhi	Dyes hydroxynaphthol blue and SYBR Green	No	Nucleic acid	4.8 × 10^4^ CFU	NR	NR	[15]
5	*Salmonella* Typhimurium	Aptamer with enzymatic signal amplification	Milk and 2 g of pork	Bacteria suspension	42 CFU/mL	NR	NR	[19]
6	*Salmonella* Typhimurium, *Salmonella* Paratyphi A	Gold nanoparticles	No	Bacteria suspension	10^2^ CFU/mL	NR	NR	[20]
7	*Salmonella* Enteritidis	Gold nanoparticles	Milk sample (spiked)	Nucleic acid	1.2 × 10^2^ CFU/mL	NR	NR	[21]
8	*Salmonella* Enteritidis	RPA	Food samples (spiked)	Nucleic acid	5 × 10^3^ CFU/mL	95%	85%	[22]
9	*Salmonella* spp.	Oligonucleotide probe	Water sample (spiked)	Nucleic acid	NR	NR	NR	[23]
10	*Salmonella* spp.	DNA probe	95 raw chicken meat, 111 animal feed, 90 poultry feces, 53 liquid egg and 30 frozen food samples	Nucleic acid	10^5^ CFU/mL	98.2%	99.5%	[24]
11	*Salmonella* Typhimurium	RPA	Hazelnut, peanut, soybean, tomato, maize	Nucleic acid	6 CFU/mL	NR	NR	[25]
12	*Salmonella* Typhimurium	Integrated rotary microfluidic system	Water or milk sample (spiked)	Nucleic acid	50 CFU	NR	NR	[26]
13	*Salmonella* spp.	Gold nanoparticles	Human DNA samples	Nucleic acid	NR	NR	NR	[27]
14	*Salmonella* Typhimurium	Gold nanoparticles	Food samples (spiked)	Nucleic acid	10^5^ CFU/mL	NR	NR	[28]
15	*Salmonella* spp.	RPA	Maize (spiked)	Nucleic acid	3 CFU/mL	NR	NR	[29]

NR: Not reported; HRP: horseradish peroxidase; TMB: 3,3′5,5′-tetramethylbenzidine; RPA: recombinase polymerase amplification; DNA: deoxyribonucleic acid; SYBR: synergy brands.

## Data Availability

Not applicable.

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
