# Peer review of "Colorimetric Approach for Nucleic Acid Salmonella spp. Detection: A Systematic Review"

_ijerph, 2022, doi:10.3390/ijerph191710570_

Round 1
Reviewer 1 Report
Review of the article entitled “ Colorimetric approach for nucleic acid Salmonella spp detection” Authors: Asma Nadia Ahmad Faris et al.
Reading the article, this reviewer thought that the authors would propose the colorimetric method as adapt for clinical use. Only reading the “Conclusion” section, this reviewer realized that the authors consider this method not satisfactory, at present: only 2 of the studies selected for review report the sensitivity and specificity of the method. The reviewer – and certainly potential readers – would wonder why the authors dedicated their valuable time to reviewing a method that they consider improper.
The role of a reviewer is to propose how the authors might improve their work. Along with this mood, I suggest that the authors give value to their potentially interesting study establishing the parameters (sensitivity and specificity) of the colorimetric method. It does not take much of their time.
This reviewer is convinced that the decision - whether accept or not an article - must be reserved for the editorial committee. However, some journals ask the reviewer to express his opinion also on this issue. In this case, the answer of this reviewer is yes: the article deserves to be published, provided that the authors accept to dedicate more time to their work and define its sensitivity and specificity.
Author Response
Thank you for your feedback.

Reviewer 2 Report
Comments and suggestions for Authors
Manuscript ID: ijerph-1826804
Type of manuscript: Review
Title: Colorimetric Approach for Nucleic Acid Salmonella spp Detection: A
Systematic Review
The paper titled, " Colorimetric Approach for Nucleic Acid Salmonella spp Detection: A Systematic Review", can be an important consideration for some investigators, but there are comments that need to be addressed by the authors
The development of rapid detection methods for Salmonella spp. and rapid identification of the source of infection by subtyping are important for the surveillance and monitoring of food-borne salmonellosis. With the manufacture of advanced instruments, we have various rapid detection methods for Salmonella. (1) Please write some information about different rapid detection methods of Salmonella in foods (also about their limitations). (2) Please write more information about problems of the colorimetric method (for example about the interference of the background of food samples in the colorimetric method, which affects its accuracy) - hence a challenge that needs to be faced. (3) There are no appropriate and adequate references to related and previous work in the manuscript. (4) Please use more references and expand the discussion chapter. (5) Please remember about Salmonella nomenclature – “Salmonella” and “S.” are italicized (of course Typhi or Typhimurium is not italicized), so for example please improve figure no 3. Remember about using dots after “spp”.
Author Response
Thank you for your feedback.

Round 2
Reviewer 1 Report
no comments